# Occurrence of Antimicrobial-Resistant Bacteria in Intestinal Contents of Wild Marine Fish in Chile

**DOI:** 10.3390/antibiotics13040332

**Published:** 2024-04-05

**Authors:** Claudio D. Miranda, Christopher Concha, Luz Hurtado, Rocío Urtubia, Rodrigo Rojas, Jaime Romero

**Affiliations:** 1Laboratorio de Patobiología Acuática, Departamento de Acuicultura, Universidad Católica del Norte, Coquimbo 1780000, Chile; christopher.concha@ucn.cl (C.C.); luz.hurtado@ucn.cl (L.H.); rocio.urtubia@ce.ucn.cl (R.U.); rrojas@ucn.cl (R.R.); 2Laboratorio de Biotecnología de los Alimentos, Instituto de Nutrición y Tecnología de los Alimentos, Universidad de Chile, Santiago 7830417, Chile; jromero@inta.uchile.cl

**Keywords:** antimicrobial resistance, marine fishes, antibacterial resistant bacteria, beta lactamases, fish bacteriology, Chile

## Abstract

Antimicrobial-resistant bacteria (ARB) from the intestinal contents of wild fish may have a relevant ecological significance and could be used as indicators of antimicrobial-resistance dissemination in natural bacterial populations in water bodies impacted by urban contamination. Thus, the occurrence of ARB in the intestinal contents of pelagic and demersal wild fishes captured in anthropogenic-impacted Coquimbo Bay in Chile was studied. Culturable counts of total and antimicrobial-resistant bacteria were determined by a spread plate method using Trypticase soy agar and R2A media, both alone and supplemented with the antimicrobials amoxicillin, streptomycin, florfenicol, oxytetracycline and ciprofloxacin, respectively. Heterotrophic plate counts of pelagic and demersal fishes ranged from 1.72 × 10^6^ CFU g^−1^ to 3.62 × 10^9^ CFU g^−1^, showing variable proportions of antimicrobial resistance. Representative antimicrobial-resistant isolates were identified by 16S rRNA gene sequencing, and isolates (74) from pelagic fishes mainly belonged to *Pseudomonas* (50.0%) and *Shewanella* (17.6%) genera, whereas isolates (68) from demersal fishes mainly belonged to *Vibrio* (33.8%) and *Pseudomonas* (26.5%) genera. Antimicrobial-resistant isolates were tested for susceptibility to 12 antimicrobials by an agar disk diffusion method, showing highest resistance to streptomycin (85.2%) and amoxicillin (64.8%), and lowest resistance to oxytetracycline (23.2%) and ciprofloxacin (0.7%). Only furazolidone and trimethoprim/sulfamethoxazole were statistically different (*p* < 0.05) in comparisons between isolates from pelagic and demersal wild fishes. Furthermore, an important number of these isolates carried plasmids (53.5%) and produced Extended-Spectrum-β-lactamases (ESBL) (16.9%), whereas the detection of Metallo–β–Lactamases and class 1-integron was rare. This study provides evidence that wild fish are important reservoirs and spreading-vehicles of ARB, carrying plasmids and producing ESBLs in Chilean marine environments.

## 1. Introduction

Considering that the selection and spread of antimicrobial-resistant bacteria is one of the greatest concerns regarding the use of antimicrobials in human and animal medicine, it is critical to advance under a One Health perspective, and with knowledge of the roles of marine fishes residing in coastal waters in the emergence and maintenance of multidrug-resistant (MDR) bacteria, as well as in acting as vehicles of the spread of molecular elements involved in antibacterial resistance and transfer [1,2,3].

Most studies dealing with antimicrobial-resistant bacteria associated with fish refer to farmed fishes [4,5,6,7], but studies of the antimicrobial-resistant bacteria composing the microbiota of fishes from non-aquaculture settings are very scarce, and mostly related to freshwater environments [8,9,10,11]. Vivant et al. [11] investigated how wild fishes inhabiting a highly urbanized river are involved in the spread of antimicrobial-resistant Enterobacterales, which was mainly isolated from the gut, by observing an important occurrence of resistance against antimicrobials belonging to the penicillin class. The authors suggested that urban rivers could be considered “hot spot environments” which promote the transfer of antimicrobial-resistant bacteria between humans and wild fish. In this trend, Mills et al. [12] demonstrated that fish gut carries higher levels of genes encoding for carbapenem resistance than do the surrounding water and sediments in a river impacted by anthropogenic activities. 

It has been declared that fish are sensitive indicators for the environmental contamination of water, and are capable of accumulating antimicrobial residues by exposure in the aquatic environment [13,14,15,16]. Considering the recognized impact of urban effluents on antimicrobial resistance in aquatic habitats, as well as the high incidence of non-treated urban effluents disposed into aquatic environments in Chile [17], studies are highly necessary to assess the spread of bacteria exhibiting antibacterial resistance, resistance which could be transmitted to marine fish species which are exposed to these water bodies and commonly captured during artisanal fishing. In this trend, Ballash et al. [18] confirmed that fishes can be effective bioindicators of surface waters contaminated with antimicrobial-resistant bacteria, antimicrobial-resistance encoding genes, and antimicrobials. 

Therefore, it was considered important to evaluate the possibility that wild fishes might behave as carriers of MDR bacteria, increasing the public health importance of these species. However, no studies have been developed to evaluate the incidence of antimicrobial-resistant bacteria and transferable molecular elements such as plasmids and integrons in wild fish species caught by artisanal fishing and destined for human consumption. Contrary to this, it is believed that the feeding and ecological habits of marine fishes could lead to differing exposures to anthropic contaminants, including antimicrobial agents, so it is expected that wild fishes exhibiting demersal habits would show higher levels of resistant bacteria, exhibiting higher taxonomic diversity and distinct MDR profiles, compared to those of pelagic habits.

Thus, the study of antimicrobial resistance in marine wild fish residing in coastal waters can help to determine whether marine wild fish could act as important reservoirs and routes-of-spread of antimicrobial-resistant bacteria, posing a public health risk. The present study was undertaken to obtain information on the occurrence of antimicrobial-resistant bacteria in the intestinal contents of wild fishes inhabiting the Coquimbo Bay in Northern Chile and to determine the antimicrobial-resistance profiles and production of ESBLs, as well as the carriage of plasmids and class 1 integrons for selected antimicrobial-resistant bacteria, carried by pelagic and demersal wild fishes.

## 2. Results and Discussion

### 2.1. Antimicrobial-Resistance Levels

A high variability of culturable counts of copiotrophic and oligotrophic bacteria from the intestinal contents of wild fish species was observed, ranging from 3.29 × 10^6^ ± 2.77 × 10^5^ CFU g^−1^ to 3.62 × 10^9^ ± 3.19 × 10^9^ CFU g^−1^ in pelagic and demersal fishes, respectively (Table 1), whereas culturable counts of oligotrophic bacteria were slightly higher than copiotrophic counts, ranging from 1.72 × 10^6^ ± 1.12 × 10^5^ CFU g^−1^ to 1.42 × 10^9^ ± 1.49 × 10^9^ CFU g^−1^ in pelagic and demersal fishes, respectively (Table 2). There were no obvious differences in antimicrobial resistance between the different fish habits, and resistance proportions for streptomycin, florfenicol and amoxicillin were higher than those for oxytetracycline and ciprofloxacin.

Percentages of antimicrobial resistance as determined using copiotrophic media were highly variable among the studied wild fish species, ranging from <0.001 to 21.41% (florfenicol), as well as from <0.001 to 1.03% (oxytetracycline), <0.001 to 8.69% (amoxicillin), <0.001 to 19.55% (streptomycin), and from <0.001 to 5.86% (ciprofloxacin) (Table 1). Otherwise, the percentages of antimicrobial resistance as determined using oligotrophic media were highly variable among the studied wild fish species, ranging from < 0.001 to 23.78% (florfenicol), as well as from <0.001 to 1.54% (oxytetracycline), <0.001 to 9.41% (amoxicillin), <0.001 to 25.37% (streptomycin), and from <0.001 to 0.001% (ciprofloxacin) (Table 2).

Overall, no important differences between the percentages of antimicrobial resistance of gut bacteria from pelagic and demersal fishes were observed. There exists only one study of antimicrobial-resistant bacteria in wild fish captured from coastal impacted waters in Chile [19]. In agreement with this study, Miranda and Zemelman [19] found similarities in the frequencies of resistant bacteria found in gut samples of demersal and pelagic fishes captured in Concepción Bay, Chile. It was previously stated that the occurrence of antimicrobial-resistant bacteria in aquatic wildlife reflects both the diet of the wildlife and the impact of human activities [20,21]. The present results demonstrate that the habits of the sampled wild fishes are not a main factor determining the occurrence of antimicrobial resistance among gut bacteria. Thus, the results suggest that antimicrobial resistance can be promoted and maintained due to factors other than the fishes’ habits, and that anthropic pollution could apparently be a main factor in increasing the observed levels of antimicrobial resistance among gut bacteria in wild fish, as previously stated [11,22].

### 2.2. Bacterial Identification

Overall, pelagic fish species showed genera compositions of antimicrobial-resistant bacteria similar to those recovered from demersal fishes, exhibiting a dominance of Pseudomonas (50.0%), Shewanella (17.6%), and Vibrio (9.5%), whereas Vibrio (33.8%), Pseudomonas (26.5%), and Aliivibrio (11.8%) were predominant among resistant bacteria in demersal fishes (Table 3). Antimicrobial-resistant isolates from intestinal contents of pelagic and demersal fishes belonged to eleven and ten different genera, respectively, and from these, six genera were detected in both groups (Appendix A). The high predominance of Pseudomonas genus contrasted with the study of Miranda and Zemelman [19], who found that the resistant bacterial population from wild fish captured at Concepción Bay located at the south of Chile mainly belonged to Enterobacteriaceae and Vibrionaceae, whereas representatives of Pseudomonadaceae were almost absent. On the contrary, Kerry et al. [23] suggested that Pseudomonads may play an important role in contributing to the background frequency of antibacterial resistance in marine environments.

### 2.3. *Antimicrobial-Resistance Patterns*

A broad range of antimicrobials covering different classes (β-lactams, carbapenems, aminoglycosides, phenicols, tetracyclines, fluoroquinolones, nitrofurans, and potentiated sulfonamides) were included in the study. Isolates from pelagic fishes showed high percentages of resistance to streptomycin (94.6%), furazolidone (70.3%), and amoxicillin (60.8%), whereas they exhibited a moderate resistance to meropenem (23.0%) and oxytetracycline (29.7%), and none of these strains showed resistance to ciprofloxacin (Figure 1). 

Among resistant isolates from demersal fishes, resistance was often associated with streptomycin (75.0%), amoxicillin (69.1%), and kanamycin (54.4%), while ciprofloxacin (1.5%), oxytetracycline (16.2%), and trimethoprim/sulfamethoxazole (16.2%) resistance was less frequent (Figure 1). Only the isolate *Pseudomonas* sp. NCIS9, recovered from *M. gayi*, showed ciprofloxacin resistance, while exhibiting simultaneous resistance to nine antimicrobials. Overall, higher percentages of resistance were observed among isolates from pelagic fish species, compared with those from demersal fishes, except for the antimicrobials amoxicillin, meropenem, and kanamycin (Figure 1). Significant differences (chi-squared test, *p* < 0.05) between resistant bacteria from pelagic and demersal wild fishes were detected only for resistance to furazolidone (70.3% and 39.7%, respectively) and trimethoprim/sulfamethoxazole (40.5% and 16.2%, respectively). 

The absence of high levels of difference in the antimicrobial-resistance profiles of isolates recovered from pelagic and demersal fishes could be a consequence of a similar level of exposure to the untreated urban effluents which are disposed of into the Coquimbo Bay. Miranda and Zemelman [19] found high levels of antimicrobial multiresistance among bacteria from intestinal contents of sampled fishes, mainly belonging to the Vibrionaceae and Enterobacteriaceae families and exhibiting important proportions of resistance to ampicillin, tetracycline, streptomycin, and nitrofurantoin. Otherwise, they found that gut antimicrobial-resistant bacteria from demersal fish species showed higher levels of multiresistance than did those from pelagic fishes, differing from the results of this study. Some aspects of the situation that could explain this difference would be that, in addition to disposal of urban effluents, Concepción Bay is subject to the intense impact of industrial discharges due to the intensive activity of several fishing and fish processing companies that produce fishmeal, while Coquimbo Bay mainly receives only urban waste. On the other hand, 20 years ago in Chile, the consumption of antimicrobials was not regulated, which favored self-medication and excessive consumption of these drugs, while currently the consumption of antimicrobials requires a medical prescription, which has significantly decreased the use of antimicrobials in the Chilean population. Thus, it is possible to infer that the levels of contamination by antimicrobial residues or antimicrobial-resistant bacteria were significantly higher in Concepción Bay than the levels currently found in Coquimbo Bay.

A high number of the isolates from the pelagic and demersal fishes exhibited multi-drug resistance (MDR), a phenotype defined as simultaneous resistance to at least three classes of antimicrobials (60 out of 74 and 41 out of 68 isolates recovered from pelagic and demersal fishes, respectively), while not showing significant differences (chi-squared, *p* < 0.05) between the two groups (Figure 2). Furthermore, similar antimicrobial-resistance indices (ARI) were observed among the antimicrobial-resistant bacteria isolated from both fish groups, which exhibited ARI values of 0.46 and 0.37 for isolates recovered from pelagic and demersal fishes, respectively. The ARI index values of both groups evidence a high probability of exposure to waters impacted with MDR bacteria or contaminated with non-treated urban effluents.

Among isolates exhibiting the highest levels of simultaneous resistance, isolates NCIF25 and NCIF12, recovered from *T. murphyi* and *M. gayi*, showed simultaneous resistance to ten antimicrobials; five isolates from the pelagic species *S. japonicus* (1), S. chiliensis (2), and *T. murphyi* (2), and seven isolates from demersal species, *M. gayi* (3) and *M. ophicephalus* (4) showed resistance against nine antimicrobials (Figure 2). Otherwise, nine isolates from pelagic fish species *S. chiliensis* (6), *S. lalandi* (1), *T. murphyi* (1), and *S. japonicus* (1), as well as two isolates from demersal fishes, recovered from *M. gayi* and *M. ophicephalus*, showed simultaneous resistance against eight antimicrobials (Figure 2). It must be noted that all of these isolates were identified as belonging to the Pseudomonas genus.

### 2.4. ESBL and MBL Detection

Extended-spectrum-β-lactamases (ESBLs) conferring resistance to the expanded-spectrum cephalosporins continue to be a worldwide major public health threat [24,25]; thus, the investigation of ESBL-carrying bacteria in aquatic environments contributes to a better understanding of the roles of these environments in their dissemination under a One Health perspective. The ESBL phenotypic detection tests permitted the detection of an important incidence of isolates, which were classified as ESBL producers (13 out of 74 isolates and 11 out of 68 isolates from pelagic and demersal fishes, respectively) (Table 4). These results suggest the high probability that ESBL-producing bacteria residing in wild fish guts could be disseminated to the surrounding environment through fish feces. To advance towards a comprehensive knowledge of this issue, the detection of bla genes encoding ESBL and their association with transferable plasmids by the ESBL-producing isolates is currently under study.

The occurrence of gut ESBL-producing bacteria in wild and commercial fish has been previously reported, but these papers have mostly referred to freshwater fish. Khurana et al. [26] demonstrated a high occurrence of ESBL among the gut microbiome of endangered Tor putitora. In addition, genes encoding extended-spectrum beta-lactamase were also found in the gut microbiome of freshwater Indian carp [27]. In another study, Bollache et al. [22] detected an important incidence of ESBL-producing, cefotaxime-resistant *E. coli* from wild fish recovered from a different French river impacted by effluents from wastewater treatment plants. It has been claimed that the isolation of ESBL-producing *E. coli* from fish sampled closest to the discharges of wastewater treatment plants (WWTPs) has highlighted the risk of human contamination [11,22,28]. 

Otherwise, only a few studies reporting ESBL-carrying bacteria from marine wild fish have been published. Among them, Sousa et al. [29] demonstrated the role of gilthead seabream (*Sparus aurata*) as carriers of SHV-12 and TEM-52 extended-spectrum betalactamases-containing Escherichia coli isolates, whereas Brahmi et al. [30] reported a high prevalence of ESBL-producing Enterobacteriaceae in wild fish from the Mediterranean Sea. Furthermore, Sellera et al. [3] reported, for the first time, the occurrence of broad-spectrum cephalosporin-resistant E. coli isolates carrying *bla*_CTX-M_ –type genes encoding for ESBLs in coastal fishes from an urban polluted area of the South Atlantic Ocean of Brazil, highlighting a potential source for the dissemination of these organisms in marine ecosystems, with additional consequences for seafood safety and quality.

Only the isolates *Pseudomonas* sp. NCIA4, and *Pseudomonas fragi* NCIO10, recovered from Pacific bonito *S. chiliensis*; *Pseudomonas* sp. NCIA26, recovered from Yellowtail amberjack *S. lalandi*; and *Shewanella* sp. NCIA61, recovered from Jack mackerel *T. murphyi* produced metallo-β-lactamases (MBLs). Thus, all of the MBL-carrying isolates were recovered from pelagic fish species, whereas none of the isolates from demersal fishes produced MBLs. Currently, there are no previous reports of MBL-carrying bacteria among gut bacteria from wild or farmed fish in Chile, and this issue must be seriously considered in future investigations because of its high clinical human relevance.

### 2.5. Integron and Plasmid Detection

In this study, only the isolates *Pseudomonas* sp. NCIA62 and *Pseudomonas* sp. NCIF20 carried a class 1 integron; these were recovered from *T. murphyi* and *M. ophicephalus*, respectively (EMBL database accession numbers PP320225 and PP321470, respectively). It must be noted that isolate NCIF20 carried two plasmids (1.5 and 50 kb), whereas in isolate NCIA62, no plasmid content was detected. In agreement with the results of this study, which has shown that integron-carrying antimicrobial-resistant bacteria from wild fishes are very rare, various previous studies have demonstrated that carriage of integrons is not usually detected among wild fish gut bacteria. 

In a recent study, Vivant et al. [11] studied Enterobacterales resistant to antibiotics, isolated from wild fish inhabiting a highly urbanized river in France, and did not detect ESBL-producing bacteria, whereas class 1 integrons were observed for only three out of one hundred and five isolates, identified as *E. coli*, *Enterobacter*, and *Raoultella* isolated from the gut microbiota of a tench *Tinca tinca*, a chub *Squalius cephalus*, and a gudgeon *Gobio gobio*, respectively. In a recent study, Bourdonnais et al. [31] found a high predominance of class 1-integron integrase intI1 in demersal flatfish samples taken near major European ports, with a higher prevalence in skin samples (45.7%), but with a low incidence in gill and gut samples (5.7%), concluding that *intI1* genes emerge as robust indicators of antimicrobial-resistance contamination in the marine environment. In another study, Barraud et al. [32] found a significant difference between wild fish, in which no integrons were detected; farmed fish, with a moderate prevalence of integrons; and diseased fish, in which integron prevalence was high. The authors proposed the combination of integrons and *Aeromonas* collected from fish as potential indicators of antimicrobial resistance and anthropic pollution in the environment. 

However, the integrons carried by wild fish gut bacteria appear to be unsuitable for monitoring the dissemination of antimicrobial resistance in marine environments with different levels of anthropogenic pollution. Furthermore, Jia et al. [33] found that *intI*1 was significantly correlated with highabundance of ARGs, suggesting that *intI*1 was an important transfer element for the spread of ARGs among bacteria in the aquatic ecosystem, but in the gut ecosystem, only few genera carried *intI1,* and positive correlation was observed between *intI1* and ARGs, which might explain the relatively low abundance of *intI*1 in the gut ecosystem, demonstrating that *intI*1 was not the main element in diffusing ARGs in guppies (*Poecilia reticulata*). On the contrary, a high number of class 1-carrying bacteria were isolated from the guts of Chilean farmed salmon under intensive drug therapy [34], suggesting that class 1 integrons are ubiquitous in bacteria associated with farmed fish under intensive drug therapy, in contrast to gut resistant bacteria recovered from wild fishes.

Otherwise, a high percentage of the antimicrobial-resistant isolates recovered from wild fish exhibited plasmid content. In this study, 40 out of 74 isolates (54.0%) from pelagic fishes, and 36 out of 68 isolates (52.9%) from demersal fishes carried up to three plasmids (Table 4), without a significant difference being found in the plasmid content of pelagic and demersal fishes (chi-squared, *p* < 0.05). 

A predominance of carriage of plasmids weighing 20, 50, and 100 kb was observed among gut resistant bacteria from pelagic (5, 10, and 10 isolates, respectively) and demersal (6, 12, and 6 isolates, respectively) fishes. The high occurrence of plasmid content among the antimicrobial-resistant bacteria associated with wild fish guts is a worrying situation which could have important public health and ecological effects, and demanding further studies to elucidate which genes are carried by these plasmids, as well as to investigate their capability to be horizontally transferred.

## 3. Materials and Methods

### 3.1. Sampling and Processing of Samples

Fish samples of eight marine fish species exhibiting pelagic or demersal habits captured at Coquimbo Bay by artisanal fishery there were considered in the study. Fish samples were obtained directly from the fishermen’s boats when arriving at the port of Coquimbo city. Demersal fish species included in the study were Snakehead king-croaker (*Menticirrhus ophicephalus*), Pacific sandperch (*Prolatilus jurgularis*), Hake (*Merluccius gayi*), and Mullet (*Pinguipes chilensis*). Pelagic fish species Yellowtail amberjack (*Seriola lalandi*), Pacific bonito (*Sarda chiliensis*), Jack mackerel (*Trachrurus murphyi*), and Pacific chub mackerel (*Scomber japonicus*) were studied. 

Fish samples were aseptically transferred to sterile plastic bags, transported on ice to the Aquatic Pathobiology Lab of the Universidad Católica del Norte, and processed within 1 h after collection. Sampled fishes were aseptically eviscerated; intestinal content samples were collected by compressing the intestinal tube with sterilized pliers, and then transferred to sterile test tubes and aseptically weighed [35]. Subsequently, homogenates of intestinal content samples were made in sterile 0.9% physiological saline (PS).

### 3.2. Culturable Bacterial Counts

Culturable counts of heterotrophic and antimicrobial-resistant bacteria were determined by a spread plate method [36], using Tryptic soy agar added to 2% NaCl (TSA2, BBL BD Becton Dickinson™, Sparks, MD, USA), and R2A agar (BBL BD Becton Dickinson™) supplemented with 2% NaCl, for copiotrophic and oligotrophic bacteria, respectively. Culturable counts of total and antimicrobial-resistant bacteria were determined using plates with both media without antimicrobials and media containing a fixed amount of one of the antimicrobials (obtained from Sigma™, Darmstadt, Germany). Plates supplemented with amoxicillin (50 μg mL^−1^), streptomycin (25 µg mL^−1^), florfenicol (30 μg mL^−1^), oxytetracycline (30 μg mL^−1^), or ciprofloxacin (10 µg mL^−1^) were used to determine antibacterial-resistant culturable bacteria, whereas plates without antimicrobials were used to determine the total culturable bacteria. Assayed antimicrobials were selected because they belong to different main antimicrobial classes, such as β-lactams (amoxicillin), aminoglycosides (streptomycin), amphenicols (florfenicol), tetracyclines (oxytetracycline), and fluoroquinolones (ciprofloxacin). Appropriate 10-fold dilutions of the homogenates were inoculated (0.1 mL aliquots) in triplicate onto agar plates. All TSA2 and RA2 plates were incubated at 22 °C for 5 days. Colony-Forming Units (CFU) were enumerated, and the bacterial numbers per gram of sample were calculated, based on two or three dilution steps [36]. 

### 3.3. Bacterial Isolates

A total of 142 antimicrobial-resistant bacteria were recovered for further analysis by selection of different-looking colonies from plates with TSA2 (BBL BD Becton Dickinson™) or R2A (BBL BD Becton Dickinson™) and containing an antimicrobial agent. From these, 74 isolates were recovered from pelagic fish species, and 68 isolates were recovered from demersal fish species, as shown in Appendix A. Bacterial isolates were purified using plates with TSA2 medium and stored at −85 °C in CryoBank^TM^ vials (Mast Diagnostica, Reinfeld, Germany) before being grown in TSA2 (BBL BD Becton Dickinson™) at 20 °C for 24–48 h prior to use. 

### 3.4. Bacterial Identification

Antimicrobial-resistant isolates were identified using bacterial 16S rRNA gene sequence analysis. For amplification of the 16S rRNA genes, isolates were cultured in Tryptic soy broth (TSB2, BBL BD Becton Dickinson™) with 2% NaCl at 22 °C for 12–24 h and centrifuged at 9000× *g* for 3 min using an Eppendorf 5415D (Eppendorf AG, Hamburg, Germany) microcentrifuge to obtain a pellet. DNA extraction was carried out using the Wizard^®^ Genomic DNA Purification commercial kit (Promega, Madison, WI, USA) following the supplier’s instructions, and the obtained DNA samples were stored at −20 °C until analysis. 

The amplification of the 16S ribosomal genes of the isolates was carried out using PCR, following the methodology described by Opazo et al. [37]. The resulting amplified PCR products were sequenced with Macrogen (Rockville, MD, USA), using the ABI PRISM 373 DNA Sequencer (Applied Biosystems, Foster City, CA, USA). The sequences were edited and matched to the Ribosomal Database Project (http://rdp.cme.msu.edu/, accessed on 5 January 2024) to identify the bacterial isolates. The partial sequences of the 16S rDNA gene belonging to each isolate were deposited in the GenBank database under the accession numbers shown in Appendix A.

### 3.5. Antimicrobial-Resistance Patterns

The isolates were tested for antimicrobial susceptibility by an agar disk diffusion method according to the Clinical and Laboratory Standards Institute (CLSI) guideline VET3-A [38], using Cation-Adjusted Mueller–Hinton agar (CAMHA) (Becton Dickinson^TM^). Disks supplied by Oxoid Ltd. (Basingstoke, Hampshire, UK) and containing the following antimicrobials were used: amoxicillin (AML, 25 µg), cefotaxime (CTX, 30 µg), cefotetan (CTT, 30 µg), meropenem (MEM, 10 µg), streptomycin (S, 10 µg), kanamycin (K, 30 µg), chloramphenicol (CM, 30 µg), florfenicol (FFC, 30 µg), oxytetracycline (OXY, 30 µg), ciprofloxacin (CIP, 5 µg), furazolidone (FR, 100 µg), and co-trimoxazole (SXT, 1.25 and 23.75 µg). Plates were incubated at 22 °C for 24 h according to CLSI guidelines [38], and isolates were determined to be resistant according to the criteria established by the CLSI [39]. 

As recommended by the CLSI guideline [40], the reference strain *Escherichia coli* ATCC 25922 was included as the internal standard in all tests. The antibacterial resistance indices (ARI) of demersal and pelagic fishes were determined according to Hinton et al. [41] using the formula ARI = y/nx, in which y was the actual number of resistance determinants recorded in a population of size n, and x was the total number of antimicrobials tested for in the susceptibility test.

### 3.6. Detection of Extended-Spectrum-β-Lactamase (ESBL) Production

Production of ESBL was screened in isolates that showed reduced susceptibility to the third generation oxyimino-cephalosporin cefotaxime, using two phenotypic methods. Isolates exhibiting an inhibition zone diameter (IZD) of ≤23 mm were considered suspected ESBL-producers [42] and were subjected to confirmatory tests. ESBL production was detected phenotypically by the Combination Disc Diffusion Test (CDDT) method, in accordance with Clinical Laboratory Standard Institute (CLSI) guidelines [43,44]. A 0.5 McFarland suspension of the test isolate was prepared and inoculated onto CAMHA plates, and disks containing cefotaxime, both alone (CTX, 30 µg) and in combination with clavulanic acid (CTL), were placed 15 mm apart; the plates were incubated at 22 °C for 24 h.

The production of ESBL was confirmed in the isolates exhibiting a ≥ 5 mm increase in the inhibition zone diameter of the CTL, as compared with the CTX disc [45]. In addition, isolates were further studied for ESBL production by using the Double Disc Synergy Test (DDST) method [46]. A 0.5 McFarland suspension of the test isolate was prepared and inoculated onto MHA plates. An amoxicillin–clavulanic acid disc (AMC, 20/10 µg) was placed at the center of the CAMHA plate inoculated with the test isolate, and a ceftazidime disc (CAZ, 30 µg), ceftriaxone disc (CRO, 30 µg), and cefotaxime disc (CTX, 30 µg) were each placed 25–30 mm apart from the center disk. 

The plates were incubated at 22 °C for 24 h. An increase or distortion towards the disc of the amoxicillin–clavulanic acid-containing disc indicated synergistic activity with clavulanic acid and was considered to be positive for ESBL production [47]. The quality control of the susceptibility assays was performed; *Klebsiella pneumoniae* ATCC 700603 (ESBL producing strain) and *E. coli* ATCC 25922 (susceptible strain) were used as positive and negative control strains for the ESBL production, respectively [48].

### 3.7. Screening for Class B Metallo-β-Lactamase (MBL) Production

Bacterial isolates showing IZD of ≤23 mm to meropenem were considered likely to produce MBL enzyme [49], and these isolates were screened for MBL production using two methods. The production of MBLs, which are characterized by inhibition by the metal chelator, Ethylene diamine tetra-acetic acid (EDTA), was investigated employing a combined disc diffusion test (CDDT) method [50]. A 0.5 McFarland suspension of the test isolate was prepared and inoculated onto CAMHA plates and disks containing meropenem, both alone (MRP, 10 µg) and in combination with EDTA (MRP EDTA, 30 µg), placed 15 mm apart; the plates were incubated at 22 °C for 24 h [51]. A ≥ 7 mm increase in the IZDs of disks of MRP + EDTA compared to the IZDs of disks of MRP phenotypically confirmed the production of MBLs [52,53,54,55]. 

A control disc containing EDTA alone was used to determine the activity of the EDTA to ensure that it had not caused false positive results by inhibiting the test isolate. In addition, isolates were further studied for MBL production using the modified Hodges test (MHT) [49,56,57]. The modified Hodges, or Cloverleaf, test was performed by aseptically swabbing CAMHA plates with *E. coli* ATCC 25922. The inoculated MH agar plates were allowed to stand for about 5 min, and then single meropenem (MRP, 10 µg) disks were aseptically placed at the center of the CAMHA plates. 

The assayed isolates (adjusted to 0.5 McFarland turbidity standards) were heavily streaked on a straight line from the edge of the meropenem disc to the edge of the plate. The plates were incubated at 22 °C for 24 h and a clover-leaf-type indentation or flattening at the intersection of the tested isolate and the *E. coli* ATCC 25922 within the zone of inhibition of the meropenem disc was considered positive for MBL production [54].

### 3.8. Plasmid Content

All antimicrobial-resistant isolates were screened for their plasmid content. Briefly, the plasmid DNA obtained was run on 1.5% agarose gel electrophoresis for plasmids less than 20 kb and 0.8% agarose gel for plasmid greater than 20 kb. All bacterial isolates were screened for their plasmid content, as previously described [5]. Briefly, plasmid DNA extraction was carried out using the Wizard^®^ Plus SV Minipreps DNA Purification System (Promega, Madison, WI, USA) according to the manufacturer’s instructions Gels were stained with GelRed^TM^ (Biotium, Hayward, CA, USA) and viewed using UV transillumination. The size was estimated via comparison with standard molecular weight markers Quick-Load^®^ 1 kb Extend DNA Ladder and known plasmid weight standards [58].

### 3.9. Detection of Class 1 Integrons

The occurrences of class 1 integrons were detected using PCR, using specific primers 5′-GCCACTGCGCCGTTACCACC-3′ (forward) and 5′-GGCCGAGCAGATCCTGCACG-3′ (reverse) for the *intI1* (class 1 integron) gene, according to previously described methodologies [59,60]. Amplification of the *intI1*-integrase gene was performed in a BIOER Technology^TM^ model GE-96G Thermal cycler using the following cycle conditions: 95 °C for 30 min, followed by 30 cycles of 95 °C for 30 s, 58 °C for 30 s (annealing temperature), and 72 °C for 1 min. There was a final extension at 72 °C for 5 min. PCR products for *intI1* gene were confirmed using restriction enzymes, considering that the SphI enzyme produces two fragments (393 and 499 bp). The obtained fragments were visualized using 1.5% agarose gel electrophoresis. Gels were stained with GelRed^TM^ (Biotium, Hayward, CA, USA) and viewed using UV transillumination. *Citrobacter gillenii* FP75 [5] was included as a positive control in all assays.

### 3.10. Statistical Analysis

The percentages of antimicrobial resistance of isolates from pelagic and demersal fishes were compared with Pearson’s chi-square test, adjusted with Bonferroni’s correction, and *p* < 0.05 was considered to indicate statistical significance. All statistical analyses were carried out using the RStudio program, version 2023.12.1.

## 4. Conclusions

The important percentage of antimicrobial-resistant bacteria observed in this study indicates a widespread bacterial resistance to antimicrobial classes relevant for human therapy within gut bacteria in commercial fish species, most probably because of disposal of untreated urban effluents in the coastal waters studied. This is the first report of the occurrence of antimicrobial-resistant bacteria carrying plasmids and integrons in wild fish in Chile; it shows remarkable levels of plasmid-carrying bacteria, whereas the occurrence of class-1 integrons was very uncommon. 

The results of this study suggest that wild fishes captured in marine coastal waters might play relevant roles as reservoirs of multidrug-resistant bacteria, able to produce extended-spectrum-β-lactamases, and prompting the need for epidemiological studies to elucidate the origin and ability of the horizontal transfer of these elements to clarify if they represent a public health risk of concern. 

Considering that antimicrobial resistance is currently agreed to be a major public health issue, it must be faced following a One Health approach; thus, additional studies are necessary in order to assess the ecological significance of the carriage of multidrug-resistant bacteria by marine wild fishes in these environments, as well as to characterize the genetic determinants involved in these resistances, in order to compare them with the genes carried by human pathogens, emphasizing the necessity of clarifying their potential dissemination to the human compartment. Otherwise, occurrences of ESBL in wild fishes could be useful as indicators of levels of urban contamination of aquatic environments where the fishes are caught, representing an useful indicator of anthropogenic pollution.

## Figures and Tables

**Figure 1 antibiotics-13-00332-f001:**
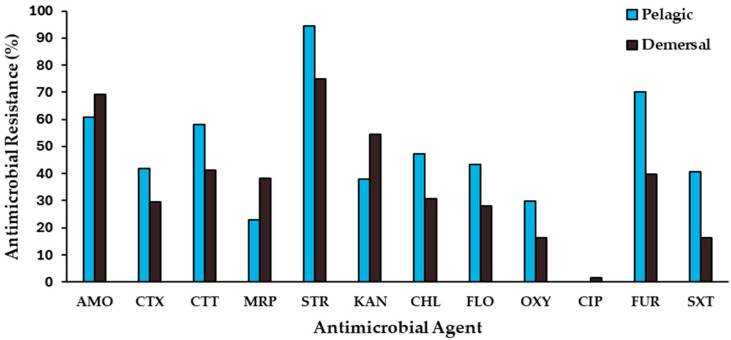
Frequency of resistance to antimicrobials of bacterial isolates recovered from intestinal contents of pelagic (n = 74) and demersal (n = 68) wild fish species from Coquimbo Bay. Antimicrobial abbreviations: AMO, Amoxicillin; CTX, Cefotaxime; CTT, Cefotetan; MRP, Meropenem; CHL, Chloramphenicol; FLO, Florfenicol; STR, Streptomycin; KAN, Kanamycin; OXY, Oxytetracycline; CIP, Ciprofloxacin; FUR, Furazolidone; SXT, Trimethoprim/Sulfamethoxazole.

**Figure 2 antibiotics-13-00332-f002:**
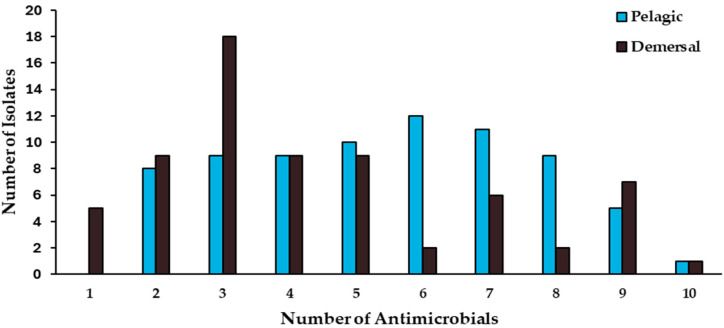
Antimicrobial multiresistance of bacterial isolates recovered from intestinal contents of pelagic (n = 74) and demersal (n = 68) wild fish species in Coquimbo Bay.

**Table 1 antibiotics-13-00332-t001:** Copiotrophic bacterial culturable count (CBCC) and antimicrobial resistance in the intestinal contents of wild fishes from Coquimbo Bay.

Habitat	Common Name	Scientific Name	CBCC ± SD (CFU g^−1^)	Antimicrobial Resistance ± SD (%)				
				FLO (30 µg mL^−1^)	OXY (30 µg mL^−1^)	AMO (50 µg mL^−1^)	STR (25 µg mL^−1^)	CIP (10 µg mL^−1^)
Pelagic	Pacific chub mackerel	*Scomber japonicus*	4.93 × 10^8^ ± 4.58 × 10^8^	21.41± 8.617	0.30 ± 0.07	8.69 ± 5.70	12.34 ± 5.02	0.002 ± 0.001
	Yellowtail amberjack	*Seriola lalandi*	5.17 × 10^7^ ± 6.13 × 10^7^	0.89 ± 1.18	0.07 ± 0.03	0.95 ± 0.09	5.86 ± 6.55	<0.001
	Jack mackerel	*Trachurus murphyi*	4.76 × 10^7^ ± 3.01 × 10^7^	0.04 ± 0.06	0.02 ± 0.01	0.40 ± 0.08	<0.001	5.86 ± 6.55
	Pacific bonito	*Sarda chiliensis*	3.29 × 10^6^ ± 2.77 × 10^5^	5.11 ± 1.61	1.03 ± 0.62	5.30 ± 1.43	19.55 ± 8.20	<0.001
Demersal	Chilean sandperch	*Pinguipes chilensis*	3.62 × 10^9^ ± 3.19 × 10^9^	<0.001	<0.001	0.34 ± 0.07	5.28 ± 3.01	<0.001
	Snackerel king-croaker	*Menticirrhus ophicephalus*	6.68 × 10^7^ ± 5.16 × 10^7^	0.17 ± 0.06	0.06 ± 0.05	1.36 ± 1.10	27.76 ± 7.29	<0.001
	Pacific sandperch	*Prolatilus jurgularis*	1.75 × 10^8^ ± 2.74 × 10^7^	<0.001	<0.001	0.80 ± 0.75	13.15 ± 15.58	<0.001
	Common hake	*Merluccius gayi*	8.43 × 10^7^ ± 4.46 × 10^7^	0.26 ± 0.05	0.01 ± 0.01	0.36 ± 0.11	0.62 ± 0.13	<0.001

FLO: Florfenicol; OXY: Oxytetracycline; AMO: Amoxicillin; STR: Streptomycin; CIP: Ciprofloxacin.

**Table 2 antibiotics-13-00332-t002:** Oligotrophic bacterial culturable count (OBCC) and antimicrobial resistance in the intestinal contents of wild fishes from Coquimbo Bay.

Habitat	Common Name	Scientific Name	CBCC ± SD (CFU g^−1^)	Antimicrobial Resistance ± SD (%)				
				FLO (30 µg mL^−1^)	OXY (30 µg mL^−1^)	AMO (50 µg mL^−1^)	STR (25 µg mL^−1^)	CIP (10 µg mL^−1^)
Pelagic	Pacific chub mackerel	*Scomber japonicus*	4.50 × 10^8^ ± 3.90 × 10^8^	23.78 ± 9.12	0.25 ± 0.03	9.34 ± 4.45	10.71 ± 4.90	0.001 ± 0.001
	Yellowtail amberjack	*Seriola lalandi*	3.38 × 10^7^ ± 4.63 × 10^7^	1.48 ± 1.23	0.27 ± 0.34	4.76 ± 5.03	12.73 ± 16.36	<0.001
	Jack mackerel	*Trachurus murphyi*	1.69 × 10^7^ ± 1.77 × 10^7^	0.10 ± 0.13	0.08 ± 0.10	0.56 ± 0.47	14.83 ± 3.97	<0.001
	Pacific bonito	*Sarda chiliensis*	1.72 × 10^6^ ± 1.12 × 10^5^	6.27 ± 0.65	1.54 ± 0.25	5.77 ± 0.58	18.87 ± 3.32	<0.001
Demersal	Chilean sandperch	*Pinguipes chilensis*	1.42 × 10^9^ ± 1.49 × 10^9^	<0.001	<0.001	0.68 ± 0.87	2,81 ± 0,85	<0.001
	Snackerel king-croaker	*Menticirrhus ophicephalus*	3.61 × 10^7^ ± 3.56 × 10^7^	0.22 ± 0.04	0.03 ± 0.02	0.42 ± 0.01	25.37 ± 11.42	<0.001
	Pacific sandperch	*Prolatilus jurgularis*	1.30 × 10^7^ ± 8.75 × 10^6^	<0.001	<0.001	9.41 ± 11.54	8.59 ± 2.17	<0.001
	Common hake	*Merluccius gayi*	2.87 × 10^6^ ± 3.29 × 10^6^	6.90 ± 0.03	0.04 ± 0.04	8.37 ± 6.09	20.67 ± 7.77	<0.001

FLO: Florfenicol; OXY: Oxytetracycline; AMO: Amoxicillin; STR: Streptomycin; CIP: Ciprofloxacin.

**Table 3 antibiotics-13-00332-t003:** Source and molecular identification of bacterial isolates.

Genus	Pelagic					Demersal					Total
	*S. lalandi*	*S. japonicus*	*S. chiliensis*	*T. murphyi*	Total	*M. gayi*	*M. ophicephalus*	*P. chilensis*	*P. jugularis*	Total	
*Acinetobacter*			1		1	1				1	2
*Aliivibrio*							4		4	8	8
*Brevibacterium*				1	1						1
*Brochothrix*		1	4	1	6						6
*Myroides*			1		1						1
*Moellerella*	1				1		1			1	2
*Photobacterium*	1				1		1		3	4	5
*Proteus*							1			1	1
*Providencia*	1				1						1
*Pseudoalteromonas*							2			2	2
*Pseudomonas*	5	7	16	9	37	11	7			18	55
*Psychrobacter*	4			1	5	2	2			4	9
*Shewanella*	4	1	2	6	13		4		2	6	19
*Vibrio*	5	2			7	2	6	9	6	23	30
**Total**	**21**	**11**	**24**	**18**	**74**	**16**	**28**	**9**	**15**	**68**	**142**

**Table 4 antibiotics-13-00332-t004:** Antimicrobial-resistance properties of bacterial isolates recovered from wild fishes.

Habitat	Fish Species	No. Isolates	MDR	ARI	Enzyme Production		Plasmid Content				Intl1
					ESBL	MBL	0	1	2	3	
Pelagic	*Seriola lalandi*	21	14	0.4	4	0	5	9	6	1	0
	*Scomber japonicus*	11	8	0.42	1	0	7	3	1	0	0
	*Sarda chiliensis*	24	22	0.51	6	2	14	7	2	1	0
	*Trachurus murphyi*	18	16	0.48	1	1	8	7	2	1	1
Demersal	*Merluccius gayi*	16	13	0.52	7	0	5	9	2	0	0
	*Menticirrhus ophicephalus*	28	14	0.36	4	0	12	10	5	1	1
	*Pinguipes chilensis*	9	6	0.26	0	0	7	2	0	0	0
	*Prolatilus jurgularis*	15	8	0.29	0	0	8	7	0	0	0
**Total**		**142**	**101**	**0.37**	**23**	**3**	**66**	**54**	**18**	**4**	**2**

ARI: Antimicrobial Resistance Index; ESBL: Extended-Spectrum-β-Lactamase; MBL: Metallo-β-Lactamase; IntI1: Class-1 integrase.

## Data Availability

The 16S RNA sequences of isolates have been deposited at DDBJ/ENA/GenBank under the accession numbers described in Appendix A.

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
