# Peer review of "Occurrence of Antimicrobial-Resistant Bacteria in Intestinal Contents of Wild Marine Fish in Chile"

_antibiotics, 2024, doi:10.3390/antibiotics13040332_

Round 1

Reviewer 1 Report

Comments and Suggestions for Authors

Occurrence of antimicrobial-resistant bacteria in intestinal content of wild marine fishes in Chile

The study investigates antimicrobial-resistant bacteria (ARB) in the intestinal contents of wild fish from Coquimbo Bay, Chile, an area affected by human activity. The study concludes that wild fish are key reservoirs and vectors for the dissemination of ARB in marine environments impacted by urban pollution. This study is both interesting and timely, investigating the impact of anthropogenic activities on the spread of ARB in the natural environment. It provides valuable insights into the extent of environmental disruption caused by human actions. When it comes to scientific names, the authors have done a good job in formatting them. However, I have noticed a few issues that need to be fixed.

Abstract

Line 16: Use "media" as it refers to more than one type of medium

Line 19-22: When referring to the genus, the singular form is "genus" and plural is "genera". Since multiple genera are mentioned, use "genera".

Introduction

The introduction appears to be a detailed literature review. It contains an excessive amount of information. Consider condensing it to provide readers with a more accessible and straightforward introduction.

Line 38: "one health perspective" should be capitalized as "One Health perspective" to reflect the recognized global strategy. Make it consistent. Some places it mentions as follows, one health perspective (Line 38), one-health perspective (Line 268) and One Health (Line 537)

Lines 36,42,43, and 118: Ensure that "antimicrobial-resistant bacteria" is consistently hyphenated or make it consistent throughout the manuscript.

Line 103: "It well known" should be "It is well known".

Results and Discussion

Overall, the results and discussion sections concisely present important findings on antimicrobial resistance in marine fish, offering insights into the complexities of resistance patterns in aquatic environments. Enhancing the clarity of statistical significance, refining the discussion to more directly address the implications of these patterns, and ensuring consistency in formatting will further strengthen the presentation and impact of the results.

Materials and Methods

The methodology section is comprehensive and informative, providing a solid basis for understanding the study's procedures. Overall, it is well written.

Line 385: Ensure that "Trypticase soy broth" is correctly named "Tryptic Soy Broth" (TSB) for consistency with common laboratory nomenclature.

Line 427: Ensure "amynoglicosides" is corrected to "aminoglycosides," and "chloramfenicol" to "chloramphenicol" for spelling accuracy.

Line 432,456,480: The Mueller-Hinton agar is sometimes spelled as "Müeller-Hinton," so choose the spelling consistent with the product used. Lines 456 and 480 have abbreviations as well (MHA). Ensure acronyms and abbreviations are defined at first use.

Line 414, 467 and 458: Ensure consistency in the description of temperatures and units “ËšC” & “°C”

Line 458: Need a space between the temperature value and the unit.

Line 459: Need a space between the numerical value and the unit.

Line 522: RStudio.

Comments on the Quality of English Language

Overall, the manuscript's English quality is fine. Minor enhancements in grammatical precision and coherence, combined with constructive feedback from peers or professional editing, will significantly boost its clarity.

Author Response

We gratefully acknowledge the valuable suggestions made by the reviewer which were all considered to improve the manuscript. The Introduction section was substantially shortened and Results and Discussion section was refined, as suggested by the reviewer. Detailed responses are attached.

Reviewer 2 Report

Comments and Suggestions for Authors

In this study, the authors researched wild marine fish to determine the antimicrobial-resistant bacteria in their intestinal contents. Please find my comments below:

Introduction: The authors provided detailed information in the introduction, but it looks wordy. I was just wondering if the authors could minimize the volume of the introduction.

Line 384: “antimicrobial-resistance bacteria” should be “antimicrobial-resistant bacteria.”

Line 384-396: Did you follow any previous studies to do culturable bacterial counts? If yes, please provide an appropriate reference.

Line 390-392: I was just wondering why you used these five antibiotics. Was there any reason? Please mention it here.

Line 434-441: Please move this information to lines 426-433. It seems like the information is repeated.

Line 478: What is the full form of EDTA? Please provide the full form of any abbreviations at their first use. Please do this throughout the manuscript.

Line 521: “p” from the “p-value” should be in italics. Please correct it throughout the manuscript.

Author Response

We gratefully acknowledge the valuable suggestions made by the reviewer which were all considered to improve the manuscript. The Introduction section was substantially shortened and additional information about experiments was added, as suggested by the reviewer. Detailed responses are attached.

Reviewer 3 Report

Comments and Suggestions for Authors

A brief summary:

The manuscript is very interesting about the global problem (AMR), and every new piece of information about it would be an important step forward. This is a comprehensive study about antimicrobial-resistant bacteria from the intestinal content of wild fish captured in anthropogenic-impacted Coquimbo Bay in Chile. Overall, the article provides scientific data that would support the assessment of AMR following the One Health approach.

Review and specific comments:

In Part 3.2, you should better describe the method (why use media without and with antibiotics) and how you chose which colonies will be used for further examinations.

In Part 3.8, you should explain the method in more depth (you have cited reference 5, but you didn't mention the primers you used anywhere, the thermal profiles, or the PCR machine).

In Part 3.9, did you also use agarose gel electrophoresis to visualize the fragments?

In Part 2.2, your results for isolated cultures were different than the results from Miranda and Zemelman. You should try to explain the reason for that. 

In part 2.3, did you consider to include results of AMR from different published studies from your country which would confirm your hypothesis about anthropic pollution?

Comments on the Quality of English Language

The way English is written is quite clear and I had no issues understanding it.

Author Response

We gratefully acknowledge the valuable suggestions made by the reviewer which were all considered to improve the manuscript. Additional information about experiments was added, and several obtained results were discussed, as suggested by the reviewer.

Unfortunately, there are no recent studies on antimicrobial resistance in Chilean coastal environments. However, a recent study demonstrated the impact of effluent from sewage treatment plants on the abundance of antimicrobial resistance genes in Chilean rivers (Bueno et al., 2020), to demonstrated the anthropic impact on waters receiving non-treated effluents.

Detailed responses are attached.
